# OpenReview forum: "Initialization Matters: Unraveling the Impact of Pre-Training on Federated Learning"
_ICLR.cc/2025/Conference — Submitted to ICLR 2025_

### Official Review · Reviewer_83mf · 2024-11-01

**Soundness:** 2
**Presentation:** 2
**Contribution:** 2
**Rating:** 6
**Confidence:** 3

**Summary:**

This is a theoretical paper. This paper provides a rigorous analysis of the impact of pre-training on FedAvg's convergence. The authors demonstrate that starting from a pre-trained model can result in fewer misaligned filters at initialization, thus producing a lower test error even when the clients are heterogeneous.

**Strengths:**

- The authors provide a rigorous theoretical analysis of the impact of pre-training on federated learning, based on a two-layer CNN and synthetic datasets.
- In Proposition 1, the authors present a decomposition of the filter weights into a signal vector and a noise vector. Extending theoretical results from a centralized setting to federated learning can be challenging due to multiple local steps and data heterogeneity. Therefore, I believe that this is a solid contribution.
- Experimental results on synthetic datasets and CIFAR-10 substantiate the theoretical findings.
- This work could be valuable in practical scenarios where the number of data points per client is small, heterogeneity across clients is high, and the image modality resembles that used in training existing centralized models.

**Weaknesses:**

**Theoretical analysis**:
- Although the authors explained the reason for choosing the 2-layer CNN and I agree that this can simplify the theoretical analysis. However, this can also  limit the broader impact of the findings.
- The definition of data heterogeneity implies a binary classification problem. This means that for a multi-class problem, we would need to cast it into multiple binary classification problems to apply the data heterogeneity measure, which is less practical.
- The assumption of the noise vector being orthogonal to the signal can hardly hold in more complicated datasets.

**Practical implications**: While I understand this is a theoretical paper, I would like to know more about its practical implications. For example:
- The paper demonstrates that a pre-trained model can lead to fewer misaligned filters at initialization and lower test error even with heterogeneous clients. However, are there any insights into determining what types of pre-trained weights are best suited for what particular tasks?
- In what situations might the theory not hold?

**Questions:**

- The definitions of alignment and misalignment in Definition 1 are very interesting. Can this be quantified, for example, in an experimental setup where the data is MNIST, the model is VGG, and the task is odd and even number classification?
- Could the authors elaborate further on Figure 4, as I do not see a clear trend over r? Additionally, the green color appears to be missing in Figures 4a and 4d, or is it on top of other colors?

---

> ### Author Response · Authors · 2024-11-22
> **Response to Reviewer 83mf [1/2]**
>
> Thank you for appreciating the rigor of our analysis, extension of Proposition 1 to the federated setting and experimental results. We address the weaknesses and questions below.
>
> **Re:Weaknesses**
> 1. > Focus on two-layer CNN can limit the broader impact of the findings
>
> The key finding in our analysis is that learning the signal (for misaligned filters) becomes more challenging as data heterogeneity increases. However, this challenge can be mitigated to a large extent by starting from a pre-trained model where most filters are already aligned with the signal. We believe this finding can be extended to deeper CNN architectures as well. Intuitively, while deeper architectures provide the capacity to learn more complex hierarchical signals, aligning a filter with the signal will also progressively become harder across layers with the alignment of the L-th layer potentially depending on the alignment of the (L−1)-th layer. In non-IID FL, this suggests that feature learning deteriorates as we move from the base layer toward the final layer. This intuition is supported by empirical evidence (see experiment in Table 1 in [a]) and explains why pre-trained initialization offers greater benefits for deeper networks compared to shallower ones. We leave a more rigorous analysis with formal guarantees as future work.
>
>
>
>
> 2. > The definition of data heterogeneity implies a binary classification problem. This means that for a multi-class problem, we would need to cast it into multiple binary classification problems to apply the data heterogeneity measure, which is less practical.
>
>
> While modeling multi-class classification as multiple binary classification is one approach, we believe there can be better alternatives by changing the data generation model and subsequent theory to explicitly incorporate multi-class classification. In essence, the definition of data heterogeneity in Eq. (1) aims to capture how far the label distribution at each client is from the the global label distribution, which assigns equal probability to each label.
>
> 3. > The assumption of the noise vector being orthogonal to the signal can hardly hold in more complicated datasets.
>
> We assume orthogonality of signal and noise vectors for simplicity of analysis. This assumption can be easily relaxed as done in [b]. Our theoretical insights will remain the same with the only difference being that we need a slightly stronger condition on the dimension of the filters (Condition C2).
>
> 4. > Are there any insights into determining what types of pre-trained weights are best suited for what particular tasks?
>
> Our analysis in Section 3.4 shows that the smaller the $\ell_2$ distance between the signal in the pre-training data $\mu^{\text{pre}}$ and the signal in the fine-tuning data $\mu$, the higher will be the alignment of the pre-trained weights. Intuitively, this suggests that pre-trained weights derived from data similar to the fine-tuning task will yield better performance. An interesting direction for future work is to see if we can use empirical measures of alignment to rank the suitability of different pre-trained models for a given task.
>
> 5. > In what situations might the theory not hold?
>
> Our theory assumes a setting where data can be decomposed into signal and noise components, which is usually applicable to vision tasks. However, this assumption is less straightforward for language tasks, where decomposing data into signal and noise is more challenging, and pre-training is typically done autoregressively on text sequences. Therefore, we believe that explaining the benefits of pre-training for language models, such as Transformers, requires a different theoretical model. We are happy to discuss this in more detail if the reviewer has other situations in mind.

---

> ### Author Response · Authors · 2024-11-22
> **Response to Reviewer 83mf [2/2]**
>
> **Re:Questions**
>
> 1. > The definitions of alignment and misalignment in Definition 1 are very interesting. Can this be quantified, for example, in an experimental setup where the data is MNIST, the model is VGG, and the task is odd and even number classification?
>
> We note that even in this simple setup we cannot explicitly characterize the signal information being learned by every filter in the VGG model and hence cannot directly use Definition 1 to measure alignment. Nonetheless we can use our empirical definition of alignment introduced in Eq. (15) to measure alignment. The results are summarized in the table below, with additional experimental details and plots provided in Appendix F 2.4.
>
> [Table 1: Results on training VGG11 on MNIST to classify odd and even digits with different initializations]
> Type of Initialization |% of Misaligned Filters| Test Accuracy after 100 Rounds|
> |--|--|--|
> |Random | $12.6 \tiny{\pm 1.1}$ | $97.54 \tiny{\pm 0.03}$|
> |Pre-trained | $5.1 \tiny{\pm 0.1}$| $98.82 \tiny{\pm 0.00}$|
>
> We observe that the percentage of misaligned filters for random initialization in this task is lower compared to our experiment on CIFAR-10 in Fig. 5 where it was around $25\\%$. Intuitively, this suggests that even random features generated by deep CNNs are sufficient to achieve reasonably good test accuracy on MNIST. Nonetheless, pre-trained initialization still achieves higher accuracy, as it results in a lower percentage of misaligned filters.
>
> 2. > Could the authors elaborate further on Figure 4, as I do not see a clear trend over r? Additionally, the green color appears to be missing in Figures 4a and 4d, or is it on top of other colors?
>
> Thank you for pointing this out. We have now modified the experiment setting and plots to make the trend over $r$ more clear. In the IID case, we see that both signal learning (Fig. 4a) and noise memorization (Fig. 4b) is similar across all $r$ and grows approximately linearly with the number of local steps. In the non-IID case however while noise memorization (Fig. 4e) grows linearly for all $r$, signal learning (Fig. 4b) saturates for misaligned filters. This in turn affects the ratio of signal learning to noise memorization which is almost constant across all $r$ in the IID case (Fig. 4e) but decays with the number of local steps in the non-IID case for misaligned filters as shown in Eq. (10). Thus Figure 4 highlights the importance of filter alignment for non-IID FL and verifies our theoretical results.
>
> ---
>
> Thank you again for your review. We are happy to answer any further questions that you may have. If your concerns have been resolved, we would be really grateful if you could consider increasing your score.
>
> **References**
>
> [a] Yu, Yaodong, et al. "TCT: Convexifying federated learning using bootstrapped neural tangent kernels." Advances in Neural Information Processing Systems 35 (2022): 30882-30897.
>
>
> [b] Kou, Yiwen, et al. "Benign overfitting in two-layer ReLU convolutional neural networks." International Conference on Machine Learning. PMLR, 2023.

---

> > ### Comment · Reviewer_83mf · 2024-11-24
> >
> > I thank the authors for the detailed explanation. I will increase my score to 6.

---

> > > ### Author Response · Authors · 2024-11-25
> > > **Thank you very much for the increased score!**
> > >
> > > Thank you so much for taking the time to go through our rebuttal and increasing your score! We truly appreciate your thoughtful feedback and are glad that our response addressed your concerns.
> > >
> > > Please let us know if you have any additional comments or suggestions, we would be happy to address them. Thank you again for your time and support!

---

### Official Review · Reviewer_2aeB · 2024-11-03

**Soundness:** 3
**Presentation:** 3
**Contribution:** 2
**Rating:** 6
**Confidence:** 3

**Summary:**

The paper studies the impact of using pre-trained models on the performance of FedAvg in a data-heterogeneous environment. More precisely, the authors analyze the performance of a 2-layer CNN, both theoretically and practically with synthetic data. They introduce the notion of aligned and misaligned filters in a CNN and show that pre-training the model produces more aligned filters, resulting in higher final accuracy compared to training without pre-trained models when applying FedAvg. They extend their practical observations by running more complex experiments on CIFAR-10 using a deeper CNN.

**Strengths:**

- The paper is clear and easy to read
- The notion of aligned/misaligned filters is interesting
- The paper present the direct harmful effect on the test error of local steps and heterogeneity when the model has misaligned filter

**Weaknesses:**

- On measuring data heterogeneity. In the paper, the authors use a data heterogeneity measurement that considers only the label distribution across clients, while data heterogeneity can encompass more than this. Indeed, even for two images of the same label, after computing the model’s prediction and the associated loss for each image, their gradients could point in completely different directions or have completely different norms. For example, two images of cats—one being a real photo and the other an animation—could make the gradient vary greatly. This divergence can complicate the task and impair federated training, even when two clients hold data with the same label. The definition of data heterogeneity presented in the paper does not take into account this aspect. In the literature, a common approach is to consider the actual local gradients; see, e.g., [A] [B]. Could the authors explain their choice of measurement of data heterogeneity?

- From Figure 5 (left plot), it appears that even if the number of misaligned filters is the same for two models (for instance, at iteration 200), we can reasonably conclude from the right plot that the test error for the orange curve may never catch up to the blue one, even with 200 more training rounds. From my understanding, this suggests that even if two models have the same number of misaligned filters at initialization (e.g., using the two models at iteration 200 as initialization parameters), they may still perform differently by the end of federated learning. Therefore, it seems to me that the presence of aligned or misaligned filters alone does not fully explain why a model benefits from pre-training in heterogeneous setting. Could the authors comment on this point?


[A] Karimireddy, Sai Praneeth, et al. "Scaffold: Stochastic controlled averaging for federated learning." International conference on machine learning. PMLR, 2020.

[B] Yuan, Xiaotong, and Ping Li. "On convergence of FedProx: Local dissimilarity invariant bounds, non-smoothness and beyond." Advances in Neural Information Processing Systems 35 (2022): 10752-10765.

**Questions:**

see comments above.

---

> ### Author Response · Authors · 2024-11-22
> **Response to Reviewer 2aeB [1/2]**
>
> Thank you for appreciating the clarity of our paper, notion of aligned/misaligned filters and results regarding the harmful effect of local steps and heterogeneity on test error. We address the weaknesses below.
>
> 1. > On measuring data heterogeneity....could the authors explain their choice of measurement of data heterogeneity?
>
> We believe the reviewer here is referring to heterogeneity in domain space, i.e., while two clients can have the same label $y$, the corresponding images or $x$ values can be quite different. While domain heterogeneity can exist in FL systems, we find that label heterogeneity is the more prevalent source of heterogeneity in FL systems. This can be seen both in real world FL datasets such as Flair[a], iNaturalist[b] as well techniques to simulate heterogeneity from centralized datasets such as restricting client data to certain labels (as done in [B]) or the Dirichlet sampling technique [c].
>
> At the same time, recent works [d] have shown that the definition of data heterogeneity appearing in [A,B] is quite pessimistic in practice since they require the difference between local and global gradients to be bounded over the entire parameter space and do not accurately reflect heterogeneity in real world FL training. In this aspect, we consider our definition of heterogeneity to be a more transparent and realistic measure of heterogeneity in FL systems.
>
> Lastly, we also conduct the following experiment to show that while heterogeneity in domain space can cause the gradients to become diverse, it does not seem to affect test performance significantly. To simulate domain heterogeneity, we consider the Office-Home dataset [e] which consists of images of $65$ objects in $4$ different domains - Art, Clipart, Product and Real World. Each domain has around $20-100$ images of every object. We split the data across $4$ clients in the following ways:
> 1) **IID**: Data across all domains is split IID across clients, i.e., each client will have images corresponding to every domain and every label.
> 2) **Domain Heterogeneity**: Each client only has images corresponding to a single domain
> 3) **Label Heterogeneity**: Data is split across clients with $\alpha = 0.1$ Dirichlet label heterogeneity, i.e, each client will have images corresponding to all domains but only certain labels.
>
> [Table 1: Results on training a randomly initialized ResNet18 using FedAvg for the 3 types of data heterogeneity. Additional details can be found in Appendix F.]
> |Type of Heterogeneity |Test Accuracy After 300 Rounds | Gradient Diversity Averaged Across 300 Rounds|
> |--|--|--|
> |None (IID)| $46.70 \pm \small{0.51}$ | $3.818 \pm \small{0.007}$ |
> |Domain| $46.64 \pm \small{0.57}$ | $4.338 \pm \small{0.010}$ |
> |Label| $39.98 \pm \small{0.26}$ | $5.315 \pm \small{0.008}$|
>
>
> The table above shows while gradient diversity in the domain heterogeneity setting is higher than in the IID case, it does not significantly affect test performance of FedAvg unlike the label heterogeneity setting. We conjecture that the impact of domain heterogeneity is mitigated due to standard pre-processing data augmentations such as rotation and cropping which have a regularizing effect of enabling clients to learn similar features across domains. Thus, this experiment shows that label heterogeneity is the more challenging form of heterogeneity in FL systems and explains why most FL works, including ours, focuses on the impact of label heterogeneity on FL training.

---

> ### Author Response · Authors · 2024-11-22
> **Response to Reviewer 2aeB [2/2]**
>
> 2. > It seems to me that the presence of aligned or misaligned filters alone does not fully explain why a model benefits from pre-training in heterogeneous setting. Could the authors comment on this point?
>
>
> Great question! Our analysis for the test performance of the CNN model at round $T$ ultimately depends on bounding the ratio of the signal learning to noise memorization across all the filters. Noise memorization is independent of initialization as we discuss in Section 4 (see Eq. (14)). For signal learning, (measured by $\\langle \\mathbf{w}_{j,r}^{(T)} , j\mathbf{\mu} \\rangle$),  we have the following decomposition
>
> $$ \langle \mathbf{w}\_{j,r}^{(T)}, j\mathbf{\mu} \rangle  =  \langle \mathbf{w}\_{j,r}^{(0)}, j\mathbf{\mu} \rangle \hspace{10pt}  +  \underbrace{\Gamma\_{j,r}^{(0, T)}}_{\text{Signal learning coefficient starting from round 0}}.$$
>
> Based on our conditions for model initialization in Section 3.3, we have that the initial correlation between the signal and filter weights at round $0$ will not be very large, i.e, $\langle \mathbf{w}\_{j,r}^{(0)} , j\mathbf{\mu} \rangle \approx 0$. Intuitively this implies that the model’s test performance begins at a near-random baseline, which is true even for pre-trained initialization. In this case we only care about the *sign* of $\langle \mathbf{w}\_{j,r}^{(0)} , j\mathbf{\mu} \rangle$, i.e. alignment, since $\Gamma\_{j,r}^{(0, T)}$ grows much faster for aligned filters as shown in Eqs. (12) and (13).
>
> On the other hand, if we consider the same analysis starting from an intermediate round $T' < T$, then
>
> $$ \langle \mathbf{w}\_{j,r}^{(T)}, j\mathbf{\mu} \rangle = \langle \mathbf{w}\_{j,r}^{(T')}, j\mathbf{\mu} \rangle \hspace{10pt} + \underbrace{\Gamma\_{j,r}^{(T', T)}}_{\text{Signal Learning Coefficient starting from round $T'$}}.$$
>
> In this case, in addition to the signal learning coefficient we also have to account for the *magnitude* of filter-signal correlation at initialization, i.e. $\langle \mathbf{w}\_{j,r}^{(T')}, j\mathbf{\mu} \rangle \gg 0$, since the model has now trained on the signal for some rounds. For large enough $T'$, while the number of aligned filters will be same for pre-trained and random initialization (growth of $\Gamma\_{j,r}^{(T', T)}$ will be similar), we can show that the initial magnitude of signal learning across all the filters i.e, $\sum\_{j,r}  \langle \mathbf{w}\_{j,r}^{(T')}, j\mathbf{\mu} \rangle$ will be much higher for pre-trained initialization since the filters get aligned faster in this case, as also seen in Figure 5 left plot. Therefore even though all the filters may be aligned at round $T'$, pre-trained initialization will continue to have a higher value of signal learning for all subsequent rounds leading to better accuracy compared to random initialization.
>
> ---
> Thank you again for your review. We are happy to answer any further questions that you may have. If your concerns have been resolved, we would be really grateful if you could consider increasing your score.
>
> **References:**
>
> [a] Song, C., Granqvist, F., & Talwar, K. (2022). Flair: Federated learning annotated image repository. Advances in Neural Information Processing Systems, 35, 37792-37805.
>
> [b] Van Horn, Grant, et al. "The inaturalist species classification and detection dataset." Proceedings of the IEEE conference on computer vision and pattern recognition. 2018.
>
> [c] Hsu, Tzu-Ming Harry, Hang Qi, and Matthew Brown. "Measuring the effects of non-identical data distribution for federated visual classification." arXiv preprint arXiv:1909.06335 (2019).
>
> [d] Wang, Jianyu, et al. "On the unreasonable effectiveness of federated averaging with heterogeneous data." Transactions of Machine Learning Research (2024).
>
> [e] Venkateswara, Hemanth, et al. "Deep hashing network for unsupervised domain adaptation." Proceedings of the IEEE conference on computer vision and pattern recognition. 2017.
>
> [f] Nguyen, John, et al. "Where to begin? On the impact of pre-training and initialization in federated learning." International Conference of Learning Representations (2023).

---

### Official Review · Reviewer_UKYU · 2024-11-03

**Soundness:** 3
**Presentation:** 3
**Contribution:** 3
**Rating:** 8
**Confidence:** 5

**Summary:**

This paper examines the effects of initialization with pre-trained models on federated learning (FL) performance, presenting theoretical bounds for test errors in federated CNNs. The analysis emphasizes how pre-trained models reduce test error by minimizing misaligned filters, which in turn mitigates the adverse effects of data heterogeneity.

**Strengths:**

- The paper provides a detailed theoretical analysis for understanding how pre-trained initialization benefits FL, focusing on the alignment of filters and the effects of data heterogeneity on signal learning versus noise memorization.

- The conclusions resonate with intuitive understanding, suggesting that pre-trained models can improve generalization by reducing harmful overfitting due to misaligned filters, which otherwise increase error in heterogeneous FL settings.

- Experimental Support: The paper includes (though limited) experiments to support the theoretical findings.

**Weaknesses:**

- The analysis relies on a simplified two-layer CNN, raising concerns about the transferability of the derived bounds to more complex architectures often used in FL. This limitation could impact the broader relevance of the findings.

- My main concern is on the unclear assumptions and rationale. Some assumptions necessary for the main theoretical results are not thoroughly justified. For instance, Condition C2 requires a “sufficiently large” dimension d, but it is unclear whether this assumption should be intuitively reasonable. Additionally, the relationship between C1 (restricted number of updates) and C6 (learning rate sufficiently small) seems contradictory, as these conditions might imply opposing constraints on the learning process.

- While the theoretical results are insightful, it lacks practical interpretations. I would like to see more discussions on how these results might apply to real-world datasets and FL setups with varying levels of data heterogeneity (which is unknown in reality).

**Questions:**

Pls see my earlier comments.

---

> ### Author Response · Authors · 2024-11-22
> **Response to Reviewer UKYU [1/3]**
>
> Thank you for appreciating our detailed theoretical analysis, intuitive understanding of results and experimental support. We address the weaknesses below.
>
> 1. > The analysis relies on a simplified two-layer CNN, raising concerns about the transferability of the derived bounds to more complex architectures
>
> The key finding in our analysis is that learning the signal (for misaligned filters) becomes more challenging as data heterogeneity increases. However, this challenge can be mitigated to a large extent by starting from a pre-trained model where the filters are already aligned with the signal. We believe this finding can be extended to deeper CNN architectures as well. Intuitively, while deeper architectures provide the capacity to learn more complex hierarchical signals, aligning a filter with the signal will also progressively become harder across layers with the alignment of the L-th layer potentially depending on the alignment of the (L−1)-th layer. In non-IID FL, this suggests that feature learning deteriorates as we move from the base layer toward the final layer. This intuition is supported by empirical evidence (see experiment in Table 1 in [a]) and explains why pre-trained initialization offers greater benefits for deeper networks compared to shallower ones. We leave a more rigorous analysis with formal guarantees as future work.
>
>
> 2. > My main concern is on the unclear assumptions and rationale
>
> The assumptions are primarily used to ensure that the model is sufficiently overparameterized, i.e., training loss can be made arbitrarily small, and that we do not begin optimization from a point where the gradient is already zero or unbounded. We provide a more intuitive reasoning behind each of the assumptions below:
>
> * *C1) Bounded number of communication rounds:*
> This is needed to ensure that the magnitude of filter weights remains bounded throughout training since they grow logarithmically with the number of updates (Theorem 3). We note that this is quite a mild condition since the max rounds can have polynomial dependence on $1/\epsilon$ where $\epsilon$ is our desired training error.
>
> * *C2) Dimension $d$ is sufficiently large:*
> This is needed to ensure that the model is sufficiently overparameterized and the training loss can be made arbitrarily small. Recall  from our data generation model in Section 2 that our input $\mathbf{x}$ consists of a signal component $\mu \in \mathbb{R}^d$ that is common across all datapoints and noise component $\xi \in \mathbb{R}^d$ that is independently drawn from $\mathcal{N}(0,\sigma_p^2 \cdot \mathbf{I})$. Having a sufficiently large $d$ ensures that the correlation between any two noise vectors, i.e. $\langle \mathbf{\xi}, \mathbf{\xi}' \rangle/ \|\| \mathbf{\xi}\|\|^2$ is not too large (Lemma 4). Otherwise if the correlation between two noise vectors is large and negative, then minimizing the loss on one data point could end up increasing the loss on another training point which complicates convergence and prevents loss from becoming arbitrarily small.
>
> * *C3) Training set size and network width is sufficiently large:*
> The condition ensures that a sufficient number of filters get activated at initialization with high probability (Lemma 6 and Lemma 7) and prevents cases where the initial gradient is zero. The condition on training set size also ensures that there are a sufficient number of datapoints with negative and positive labels (Lemma 8).
>
> * *C4) Standard deviation of Gaussian random initialization is sufficiently small:* This condition is needed to ensure that the magnitude of the initial correlation between the filter weights and the signal and noise components, i.e $|\langle \mathbf{w}\_{j,r}^{(0)}, \mathbf{\mu} \rangle|$, $|\langle \mathbf{w}\_{j,r}^{(0)}, \xi \rangle|$ is not too large. This simplifies the analysis and prevents cases where none of the filters get activated at initialization (Lemma 21). It also ensures that after some number of rounds all filters get aligned with the signal (Lemma 30).
>
> * *C5) Norm of signal is larger than noise variance:* This condition is needed to ensure that all misaligned filters at initialization eventually become aligned with the signal after some rounds (Lemma 30). This allows us to derive a meaningful bound on test performance that is not dominated by noise memorization.

---

> ### Author Response · Authors · 2024-11-22
> **Response to Reviewer UKYU [2/3]**
>
> * *C6) Learning rate is sufficiently small:* This is a standard condition to ensure that gradient descent does not diverge. The conditions are derived from ensuring that the signal and noise coefficients remain bounded in the first stage of training and that the loss decreases monotonically in every round in the second stage of training.
>
> We have added this discussion to Appendix C where we discuss our proof sketch. We note that all these assumptions have appeared in previous work (references Cao et al. (2022); Kou et al. (2023) in the paper) and are standard in the analysis of two-layer CNN. Finally, we also clarify that these conditions do not contradict each other and can be simultaneously satisfied. The relationship between C1 (restricted number of updates) and C6 (small learning rate) is not contradictory since C1 has a factor of $1/\eta$ in its definition. So if we make the learning rate $\eta$ smaller, we are also allowed a larger number of updates.
>
> 3. > I would like to see more discussion on how results might apply to real-world datasets and FL setups with varying levels of data heterogeneity
>
>
> Thank you for the suggestion. We have added more experiments on real-world datasets and FL setups with varying levels of data heterogeneity as we discuss below.
>
> **Experiments on real-world datasets:**
>
> We extend the experiment from Figure 5 of our paper, originally conducted on CIFAR-10, to evaluate the number of misaligned filters at initialization, on more challenging and real-world datasets which include
>
> a) **TinyImageNet [b]:** $100k$ datapoints, $200$ classes, data partitioned across $20$ clients with $\alpha = 0.3$ heterogeneity
>
> b) **Google Landmarks v2 23k [c]:** $23k$ datapoints, $203$ classes, $233$ clients, data naturally grouped by photographer to achieve a federated partitioning
>
> The results are summarized in the table below, with additional experimental details and plots provided in Appendix F 2.1.
>
> [Table 1: Results on training ResNet-18 with FedAvg across various datasets]
> |Dataset |Type of Initialization |% of Misaligned Filters| Test Accuracy After 300 Rounds| Improvement In Test Accuracy|
> |--|--|--|--|--|
> |CIFAR-10| Random | $25.5 \tiny{\pm 0.4}$ | $75.77 \tiny{\pm 0.29}$| |
> || Pre-trained | $10.6 \tiny{\pm 0.3}$| $78.58 \tiny{\pm 0.23}$| $+2.81$|
> |TinyImageNet| Random | $35.7 \tiny{\pm 0.8}$ | $36.82 \tiny{\pm0.35}$| |
> || Pre-trained | $15.0 \tiny{\pm 0.1}$| $52.63 \tiny{\pm 0.16}$| $+15.81$|
> |Google Landmarks v2| Random | $40.5 \tiny {\pm 0.7}$ | $24.65 \tiny{\pm 1.44}$|
> || Pre-trained | $12.0 \tiny{\pm 0.1}$ |$59.26 \tiny {\pm 0.87}$| $+34.61$|
>
> Observe that as the complexity of the signal in the data increases (CIFAR-10 < TinyImageNet < Google Landmarks), we see a corresponding sharp increase in the percentage of misaligned filters ($25\\%$ to $40\\%$). In contrast, with pre-trained initialization, the percentage of misaligned filters remains less than $15\\%$ across datasets leading to a larger improvement in test accuracy for harder datasets. Thus results align well with our theoretical findings: as the percentage of misaligned filters increases, the benefits of pre-training become more pronounced.

---

> ### Author Response · Authors · 2024-11-22
> **Response to Reviewer UKYU [3/3]**
>
> **Experiments on FL setup with varying levels of data heterogeneity:**
>
> We extend the experiment from Figure 5 of our paper, originally conducted on CIFAR-10 with $\alpha = 0.1$ Dirichlet heterogeneity to three other levels of heterogeneity: $\alpha = 0.05$ (high heterogeneity), $\alpha = 0.3$ (medium heterogeneity) and $\alpha = 10$ (low heterogeneity). The results are summarized in the table below, with additional experimental details and plots provided in Appendix F 2.2.
>
> [Table 2: Results on training ResNet-18 with FedAvg with different heterogeneities on CIFAR-10]
> |Level of Heterogeneity |Type of Initialization |% of Misaligned Filters |Test Accuracy After 300 Rounds|Improvement In Test Accuracy|
> |--|--|--|--|--|
> |Low ($\alpha = 10$)| Random | $24.1 \pm \small{0.9}$| $80.54 \pm \small{0.18}$||
> || Pre-trained | $9.2 \pm \small{0.1}$ |$81.94 \pm \small{0.12}$|$+1.40$|
> |Medium ($\alpha = 0.3$)| Random | $27.5 \pm \small{0.4}$ |  $75.77 \pm \small{0.29}$||
> || Pre-trained | $10.6 \pm \small{0.3}$| $78.58 \pm \small{0.23}$|$+2.81$|
> |High ($\alpha = 0.05$)| Random | $26.1 \pm \small{0.8}$ |  $64.45 \pm \small{0.10}$| |
> || Pre-trained | $11.5 \pm \small{0.3}$| $70.84 \pm \small{0.45}$|$+6.39$
>
> First, we observe that the percentage of misaligned filters remains approximately $25\\%$ with random initialization and $10\\%$ with pre-trained initialization, regardless of the level of heterogeneity. However, as heterogeneity increases, the improvement in test accuracy provided by pre-trained initialization becomes more pronounced. This trend is consistent with our theoretical analysis in Theorem 2, which suggests that the percentage of misaligned filters will have a greater impact on test performance as data heterogeneity increases.
>
>
> ---
> Thank you again for your review. We are happy to answer any further questions that you may have. If your concerns have been resolved, we would be really grateful if you could consider increasing your score.
>
> **References**
>
> [a] Yu, Yaodong, et al. "TCT: Convexifying federated learning using bootstrapped neural tangent kernels." Advances in Neural Information Processing Systems 35 (2022): 30882-30897.
>
> [b] Le, Ya, and Xuan Yang. "Tiny imagenet visual recognition challenge." CS 231N 7.7 (2015): 3.
>
> [c] Weyand, Tobias, et al. "Google landmarks dataset v2-a large-scale benchmark for instance-level recognition and retrieval."
> Proceedings of the IEEE/CVF conference on computer vision and pattern recognition. 2020.
>
> [d] Karimireddy, Sai Praneeth, et al. "Scaffold: Stochastic controlled averaging for federated learning." International Conference on Machine Learning. PMLR, 2020.
>
> [e] Wang, Jianyu, et al. "Tackling the objective inconsistency problem in heterogeneous federated optimization." Advances in Neural Information Processing Systems 33 (2020): 7611-7623.

---

> ### Comment · Reviewer_UKYU · 2024-11-24
> **Reply to the authors**
>
> I would like to thank the authors for the detailed response. I have raised my positive score from 6 to 8.

---

> > ### Author Response · Authors · 2024-11-25
> > **Thank you very much for the increased score!**
> >
> > Thank you so much for taking the time to go through our rebuttal and increasing your score! We truly appreciate your thoughtful feedback and are glad that our response addressed your concerns.
> >
> > Please let us know if you have any additional comments or suggestions, we would be happy to address them. Thank you again for your time and support!

---

### Official Review · Reviewer_Qxow · 2024-11-03

**Soundness:** 3
**Presentation:** 3
**Contribution:** 2
**Rating:** 8
**Confidence:** 3

**Summary:**

This paper addresses the question: "Why does pre-trained initialization significantly alleviate the challenges posed by non-IID data in federated learning (FL)?" To explore this, the authors first identify that the reduction in test accuracy observed in non-IID FL compared to IID FL is due to filter misalignment at initialization. They argue that when FL training begins with pre-trained models, most filters are already aligned with the signal, which mitigates the impact of data heterogeneity. The paper is primarily theoretical, with less emphasis on experimental validation.

**Strengths:**

1. The paper is technically well-written and easy to follow, with well-chosen and consistent notation throughout.
2. The results presented in Section 4 are intriguing and, to the best of my knowledge, novel contributions to the field.

**Weaknesses:**

1. I understand that the results derived for the two-layer CNN can, to some extent, be generalized to deeper CNN architectures. However, I would appreciate if the authors could discuss how the theorems introduced in the paper might change under such a generalization.

2. While the paper is primarily theoretical, additional experiments on larger datasets would significantly strengthen the paper’s contributions.

3. In line 357, the authors mention: "We focus on centralized pre-training, but our discussion here can be extended to federated pre-training as well." It is unclear why this extension holds. The authors should elaborate on this claim elsewhere in the paper. Overall, the primary concern with this manuscript is that it shifts between centralized and FL settings without clear distinction.

4. Minor issues:

   a) The authors inconsistently capitalize the first letters when introducing abbreviations, e.g., "Independent and Identically Distributed (IID)" versus "machine learning (ML)." Please ensure consistency throughout the paper.

   b) On lines 54-56, the authors mention, "One reason suggested by Nguyen et al. (2022) is a lower value of the training loss at initialization when starting from pre-trained models." Could the authors clarify if this statement refers to the centralized or FL setup?

   c) Line 71: "…two-layer ReLU convolutional neural networks (CNNs) (Zou et al., 2023)..." The current citation format implies that the two-layer ReLU architecture was introduced by (Zou et al., 2023). A clearer phrasing might be: "similar to (Zou et al., 2023), we use a two-layer ReLU..."

   d) Line 116: "Also, [a] denotes {1, 2, . . . , n}." Please correct this typo for clarity.

**Questions:**

Please address the concerns I raised above, and explicitly clarify whether your theorems apply to centralized settings, federated learning (FL) settings, or both. Provide clear reasoning to support each case.

---

> ### Author Response · Authors · 2024-11-22
> **Response to Reviewer Qxow [1/2]**
>
> Thank you for appreciating the clarity of our writing and novel contributions. We address the weaknesses and questions below.
>
> 1. > I would appreciate if the authors could discuss how the theorems introduced in the paper might change under generalization to deeper CNN architectures
>
> The key finding in our analysis is that learning the signal (for misaligned filters) becomes more challenging as data heterogeneity increases. However, this challenge can be mitigated to a large extent by starting from a pre-trained model where the filters are already aligned with the signal. We believe this finding can be extended to deeper CNN architectures as well. Intuitively, while deeper architectures provide the capacity to learn more complex hierarchical signals, aligning a filter with the signal will also progressively become harder across layers with the alignment of the L-th layer potentially depending on the alignment of the (L−1)-th layer. In non-IID FL, this suggests that feature learning deteriorates as we move from the base layer toward the final layer. This intuition is supported by empirical evidence (see experiment in Table 1 in [a]) and explains why pre-trained initialization offers greater benefits for deeper networks compared to shallower ones. We leave a more rigorous analysis with formal guarantees as future work.
>
>
> 2. > Additional experiments on larger datasets would significantly strengthen the paper’s contributions.
>
> We extend the experiment from Figure 5 of our paper, originally conducted on CIFAR-10, to evaluate the number of misaligned filters at initialization, on more challenging and real-world datasets which include
>
> a) **TinyImageNet [b]:** $100k$ datapoints, $200$ classes, data partitioned across $20$ clients with $\alpha = 0.3$ heterogeneity
>
> b) **Google Landmarks v2 23k [b]:** $23k$ datapoints, $203$ classes, $233$ clients, data naturally grouped by photographer to achieve a federated partitioning
>
> The results are summarized in the table below, with additional experimental details and plots provided in Appendix F 2.1.
>
> [Table 1: Results on training ResNet-18 with FedAvg across various datasets]
> |Dataset |Type of Initialization |% of Misaligned Filters| Test Accuracy After 300 Rounds| Improvement In Test Accuracy|
> |--|--|--|--|--|
> |CIFAR-10| Random | $25.5 \tiny{\pm 0.4}$ | $75.77 \tiny{\pm 0.29}$| |
> || Pre-trained | $10.6 \tiny{\pm 0.3}$| $78.58 \tiny{\pm 0.23}$| $+2.81$|
> |TinyImageNet| Random | $35.7 \tiny{\pm 0.8}$ | $36.82 \tiny{\pm0.35}$| |
> || Pre-trained | $15.0 \tiny{\pm 0.1}$| $52.63 \tiny{\pm 0.16}$| $+15.81$|
> |Google Landmarks v2| Random | $40.5 \tiny {\pm 0.7}$ | $24.65 \tiny{\pm 1.44}$|
> || Pre-trained | $12.0 \tiny{\pm 0.1}$ |$59.26 \tiny {\pm 0.87}$| $+34.61$|
>
> Observe that as the complexity of the signal in the data increases (CIFAR-10 < TinyImagenet < Google Landmarks), we see corresponding a sharp increase in the percentage of misaligned filters ($25\\%$ to $40\\%$). In contrast, with pre-trained initialization, the percentage of misaligned filters remains less than $15\\%$ across datasets leading to a larger improvement in test accuracy for harder datasets. These results align well with our theoretical findings: as the percentage of misaligned filters increases, the benefits of pre-training become more pronounced.

---

> ### Author Response · Authors · 2024-11-22
> **Response to Reviewer Qxow [2/2]**
>
> 3. > Explicitly clarify whether your theorems apply to centralized settings, federated learning (FL) settings, or both. Provide clear reasoning to support each case.
>
> Theorems 1 and 2 are applicable to both centralized and federated learning (FL) settings, as the centralized setting can be viewed as a special case of FL where clients perform only a single local step (i.e., $\tau = 1$). However, the effects of data heterogeneity and misaligned filters on the test error bound in Theorem 2 become evident only when $\tau > 1$, which is the typical scenario in FL. Thus Theorem 2 highlights that initialization plays a more critical role in federated settings compared to centralized ones, as also seen in practice.
>
> 4. > In line 357, the authors mention: "We focus on centralized pre-training, but our discussion here can be extended to federated pre-training as well." It is unclear why this extension holds.
>
>
> The key insight in our discussion is that misaligned filters eventually align with the signal in both centralized and FL settings. In FL, however, the number of rounds taken for all the filters to get aligned will increase as the data heterogeneity increases. This is demonstrated empirically by our new experiments in Appendix F 2.2 where we extend the experiment from Figure 5 of our paper, originally conducted on CIFAR-10 with $\alpha = 0.1$ Dirichlet heterogeneity to three other levels of heterogeneity: $\alpha = 0.05$ (high heterogenity), $\alpha = 0.3$ (medium heterogeneity) and $\alpha = 10$ (low heterogeneity).
>
> Since since pre-training is usually done on a public dataset, we focus on centralized pre-training in the discussion in Section 3.4. As such, Lemma 2 gives an upper bound on the number of iterations needed for all filters to get aligned in the centralized setting. As part of our proof for Theorem 2 we also derive the federated counterpart of this lemma- Lemma 30 in Appendix B - which gives an upper bound on the the number of rounds required for all filters to get aligned in the FL setting. We have now added an explanation after line 357 to clarify this extension.
>
> 5. > Minor Issues
>
> Thank you for the careful proofreading! We have fixed the points mentioned in points (a), (c), (d). We also clarify that the statement in Nguyen (2022) refers to an FL setup.
>
> ---
> Thank you again for your review. We are happy to answer any further questions that you may have.
>
> **References**
>
> [a] Yu, Yaodong, et al. "TCT: Convexifying federated learning using bootstrapped neural tangent kernels." Advances in Neural Information Processing Systems 35 (2022): 30882-30897.
>
> [b] Le, Ya, and Xuan Yang. "Tiny imagenet visual recognition challenge." CS 231N 7.7 (2015): 3.
>
> [c] Weyand, Tobias, et al. "Google landmarks dataset v2-a large-scale benchmark for instance-level recognition and retrieval." Proceedings of the IEEE/CVF conference on computer vision and pattern recognition. 2020.

---

> > ### Comment · Reviewer_Qxow · 2024-11-23
> > **Reply to the authors**
> >
> > I would like to thank the authors for the detailed response. My concerns have been mainly addressed, and thus I will keep my positive score.

---

> > > ### Author Response · Authors · 2024-11-25
> > > **Thank you so much for the positive score!**
> > >
> > > Thank you so much for taking the time to go through our rebuttal! We truly appreciate your thoughtful feedback and are glad that our response addressed your concerns.
> > >
> > >
> > > Please let us know if you have any additional comments or suggestions, we would be happy to address them. Thank you again for your time and support!

---

### Public Comment · ~Divyansh_Jhunjhunwala1 · 2025-02-14
**Response to Meta Review [1/2]**

First and foremost, as authors of this paper, we appreciate the time and effort that went into reviewing our submission. At the same time, we are disappointed by the AC's decision to reject our paper despite all reviewers unanimously recommending acceptance (scores: 8, 8, 6, 6).  The meta-review raises concerns that we believe misrepresents our contributions and is inconsistent with prior work. Moreover, several of these concerns were introduced for the first time in the meta-review, giving us no opportunity to address them during the rebuttal. It is also unclear whether the AC meaningfully engaged with the reviewers before assuming they overlooked key aspects and overturning their recommendations.

Below, we address the key points raised in the meta-review.

**1. Extension to deeper CNNs:** As pointed out in the rebuttal, given a signal-noise data model, the main adjustment to incorporate new model architectures would be in Lemma 1, which quantifies how the signal and noise coefficients of the model weights evolve during training. Intuitively, we expect the growth of the noise coefficients to be relatively unaffected by the initialization (both random and pre-trained initialization can overfit the data) and the signal coefficients to grow faster if the model weights are already aligned with the signal. Section 4 formalizes this intuition for the two-layer CNN and our main result in Theorem 2 shows how the test error of the model depends on the ratio of signal learning to noise memorization across the filters. *Thus, as long as we can establish that the signal and noise coefficients grow at different rates (depending on initialization, data heterogeneity), the core insights in our work can be extended to other architectures.*

However, we emphasize that we do not claim this extension to be trivial or straightforward. Even for a two-layer CNN, analyzing the growth of signal and noise coefficients requires significant theoretical effort. Theoretical progress happens in steps, and our results take the first step towards a more rigorous understanding of why pre-training benefits FL.


**2. Pre-assignment of filters to classes:** Note that the assignment of a filter to a class is just determined by the sign of the weight connected to that filter in the second layer (which we assume to be fixed). Furthermore it is empirically well documented that filters in the deeper layers of a CNN learn class-specific features [1,2,3]. Our model simply formalizes this structure for theoretical analysis.

**3. Gap between theoretical results and experiments**: Our introduction explicitly states:

> We experimentally verify our upper bound on test error in a synthetic data setting (Section 3) and further validate our insights on practical FL tasks with deeper CNNs."

This is a standard approach in theoretical ML research: first, verifying theoretical predictions in controlled settings, and second, demonstrating that the insights generalize to practical scenarios. Many impactful recent works [4,5,6,7] adopt this methodology, and it is unclear why our work is being singled out for criticism. The meta-review's characterization of this gap as a major issue is surprising.

**4. Data model:** The assumption that noise vectors follow a zero-mean Gaussian distribution is a widely accepted model spanning across ML, statistics and signal processing literature [8, 9] owing to its mathematical tractability. We also note that the normal distribution with singular covariance matrix can be defined by focusing on the subset of coordinates where the singular value is positive (see Definition 2.4.1 in [10])

**5. Extension to ViTs:** We want to reiterate that we do not claim our theoretical results can be applied to ViTs *straightforwardly*. Our goal during the rebuttal was, first, to clarify how the definition of alignment can be extended to ViTs and, second, to outline a general approach for adapting our analysis to this setting. In response to the AC’s questions, we explicitly detailed how the convention of convolutional filters in CNNs can be extended to the query, key, and value matrices in ViTs (https://openreview.net/forum?id=GYik1jT3gE&noteId=IWu0G5Kzy5). To paraphrase our discussion:

> Each column of the Q, K, and V matrices can be viewed as analogous to filters in our CNN model.

Our empirical results further demonstrate that the attention weights in ViTs exhibit strong alignment with signal information, reinforcing the core intuition behind our analysis. Furthermore, while ViTs are undoubtedly an important architecture, the emphasis placed on them in the decision process is not justified. Understanding the impact of pre-training in federated learning remains an open problem—even for CNNs—and our work provides foundational insights into this broader challenge.

---

---

> ### Public Comment · ~Divyansh_Jhunjhunwala1 · 2025-02-14
> **Response to Meta Review [2/2]**
>
> We also take this opportunity to respond to several claims made by the AC in the Additional Discussion:
>
> **Claim 1:**
>
> >"Most reviews of this paper are relatively light."
>
> We are not sure how the AC comes to this conclusion, as each review provides detailed strengths, weaknesses, and constructive suggestions for improvement. If the AC genuinely believed the reviews were insufficient, we would have hoped the AC could solicit additional experts to review and foster proper discussion among reviewers.
>
> **Claim 2:**
>
> > ...analysis here is specific to a superficial CNN model
>
> We disagree with the claim that the two-layer CNN model is "superficial." While it is a simplification of a standard CNN model, it has been widely used in theoretical ML research to gain insights into deep networks. Several recent works have demonstrated that the two-layer CNN model captures key phenomena observed in practical settings, including:
> * The generalization gap between Adam and SGD [11]
> * The emergence of benign overfitting [12,13]
> * The benefits of FedAvg over local training [14]
> * The effectiveness of data augmentation techniques like Cutout and CutMix [15]
> * The role of local steps in FL for learning heterogeneous features [16]
>
> By adopting the two-layer CNN and corresponding data model, we eliminate the need for common assumptions in FL convergence, such as smoothness and bounded heterogeneity [17, 18]. Instead, this setup allows us to introduce a more interpretable notion of heterogeneity based on label proportion mismatch across clients (Equation 1), establish convergence to a global minimum for a non-convex problem and provide generalization guarantees.
>
> **Claim 3:**
>
> > "No reviewer has mentioned this important point in the review, so I suspect that the reviewers have not read the theoretical setup and analysis in detail."
>
> This assertion is unfounded and unfair. A lack of concern from the reviewers does not imply that they did not carefully read the paper.  We would have hoped that the AC could consult the reviewers to verify this before overriding their decision.
>
>
>
> ---
> In conclusion, we sincerely appreciate the reviewers' thoughtful evaluations and their unanimous recognition of the novelty and significance of our work. However, as outlined above, we are disappointed by the inconsistencies and claims made in the AC’s assessment without sufficient justification. We hope our response fosters a broader discussion on the fairness and transparency in the ML review process.
>
> Best,
>
> Divyansh, Pranay, Zheng, Gauri

---

> > ### Public Comment · ~Divyansh_Jhunjhunwala1 · 2025-02-14
> > **References**
> >
> > [1] Yosinski, Jason, et al. "Understanding neural networks through deep visualization." arXiv preprint arXiv:1506.06579 (2015).
> >
> > [2] Zeiler, Matthew D., and Rob Fergus. "Visualizing and understanding convolutional networks." Computer Vision–ECCV 2014: 13th European Conference, Zurich, Switzerland, September 6-12, 2014, Proceedings, Part I 13. Springer International Publishing, 2014.
> >
> > [3] Bau, David, et al. "Network dissection: Quantifying interpretability of deep visual representations." Proceedings of the IEEE Conference on Computer Vision and Pattern Recognition. 2017.
> >
> > [4] Li, Tian, et al. "Federated optimization in heterogeneous networks." Proceedings of Machine learning and Systems 2 (2020): 429-450.
> >
> >
> > [5] Kleinberg, Bobby, Yuanzhi Li, and Yang Yuan. "An alternative view: When does SGD escape local minima?." International Conference on Machine Learning. PMLR, 2018.
> >
> > [6] Jacot, Arthur, Franck Gabriel, and Clément Hongler. "Neural tangent kernel: Convergence and generalization in neural networks." Advances in Neural Information Processing Systems 31 (2018).
> >
> > [7] Belkin, Mikhail, et al. "Reconciling modern machine-learning practice and the classical bias–variance trade-off." Proceedings of the National Academy of Sciences 116.32 (2019): 15849-15854.
> >
> >
> > [8] Hastie, Trevor, Robert Tibshirani, and Jerome Friedman. The Elements of Statistical Learning: Data Mining, Inference, and Prediction. 2nd ed., Springer, 2009.
> >
> > [9] Welch, G. "An Introduction to the Kalman Filter." (1995).
> >
> > [10] Anderson, Theodore Wilbur, et al. An introduction to multivariate statistical analysis. Vol. 2. New York: Wiley, 1958.
> >
> >
> > [11] Zou, Difan, et al. "Understanding the generalization of adam in learning neural networks with proper regularization." International Conference of Learning Representations (2023).
> >
> > [12] Cao, Yuan, et al. "Benign overfitting in two-layer convolutional neural networks." Advances in Neural Information Processing Systems 35 (2022): 25237-25250.
> >
> > [13] Kou, Yiwen, et al. "Benign overfitting in two-layer ReLU convolutional neural networks." International Conference on Machine Learning. PMLR, 2023.
> >
> > [14] Huang, Wei, et al. "Understanding convergence and generalization in federated learning through feature learning theory." International Conference on Learning Representations (2023).
> >
> > [15] Oh, Junsoo, and Chulhee Yun. "Provable benefit of cutout and cutmix for feature learning." arXiv preprint arXiv:2410.23672 (2024).
> >
> > [16] Bao, Yajie, Michael Crawshaw, and Mingrui Liu. "Provable Benefits of Local Steps in Heterogeneous Federated Learning for Neural Networks: A Feature Learning Perspective." International Conference on Machine Learning (2024).
> >
> > [17] Li, Xiang, et al. "On the convergence of fedavg on non-iid data." International Conference of Learning Representations (2020).
> >
> > [18] Wang, Jianyu, et al. "Tackling the objective inconsistency problem in heterogeneous federated optimization." Advances in Neural Information Processing Systems 33 (2020): 7611-7623.

---

### Meta-Review · Area_Chair_km5s · 2024-12-19

**Metareview:**

Despite the overall positive ratings from the reviewers, there are several critical issues with this paper:

First, unlike the authors’ claim that the theoretical analysis applies broadly to CNN models and beyond, I find it specific to a 2-layer CNN structure for binary classification. In this specific structure, the CNN filters in each layer are (as far as the theory goes) are pre-assigned to either a positive or a negative class. This is evident in most of the lemmas and definitions. See for example Definition 1; as well as the text in lines 144-148; see also Eq. (4) etc.

Second, it seems there is also a gap between the theoretical analysis and experiments of this paper. The experiments are designed for multi-class classification tasks while the theoretical analysis (as pointed out above) was designed for binary classification. Does this mean that the model used in the experiment is not the model used in the theoretical analysis?

Third, it is not clear whether the assumption regarding the data generation model is reasonable. Although I agree that for a specific task, only some image patches would be related to its class while the rest could be irrelevant, I am not sure why the latter is assumed to follow a zero-mean Gaussian. An additional (minor) concern regarding this setup is that the covariance matrix of the Gaussian might be mathematically problematic. Following the definition in line 125, it is a scaled version of I - uu^T where u = mu / ||mu|| which is a normalized vector of length 1. How do we guarantee that this is always positive definite (PD)? I think it is not PD when u approaches a one-hot vector.

—

In addition, in response to the concern regarding not conforming to state-of-the-art practice of using ViT as pre-trained backbone, the authors stated that the theoretical analysis can be extended to ViT straightforwardly. But, I am not sure if this claim holds given my first point above. As the theoretical analysis was specific to a particular CNN structure which, apparently, assigns convolutional filters in each block to either the positive or negative class, it is unclear how this convention will extend to ViT’s case with key, query, and value matrices in self-attention. Which one will be assigned with which class in this case?

—

Overall, the key concerns here are (1) it is unclear (even after rebuttal) whether the theoretical analysis, which was specifically designed for a particular CNN-based binary classifier, can be extended trivially to other CNN structures; (2) it is unclear how the theoretical analysis for binary classification can be aligned with and used to explain the experiment results in multi-class classification cases; and (3) it is also unclear how the assumption regarding the data generation model can be aligned with the datasets used in the experiments?

As such, I feel that the amount of clarifications required for these technical ambiguities are extensive and will require a significant revision of the paper. Therefore, I cannot recommend acceptance for this paper in its current form.

**Additional Comments On Reviewer Discussion:**

Overall, most reviews of this paper are relatively light. Most of the reviews did not seem to raise concerns regarding the disparity between the theoretical and empirical setups (2-class vs multi-class classification). One reviewer did point this out but it seems the reviewer viewed it as a minor point. However, I cannot agree that this is a minor point because as it stands, the theoretical and empirical analyses have been misaligned.

Furthermore, it is also concerning that the theoretical analysis here is specific to a superficial CNN model that pre-assigns each filter to one of the (2) classes in a 2-class classification setup. No reviewer has mentioned this important point in the review so I suspect that the reviewers have not read the theoretical setup and analysis in detail, and have missed it. It is therefore unclear how the develop theories will extend to ViT as claimed by the authors.

Given this, I do not think there is a solid basis for accepting this paper. Due to the multiple ambiguities that I pointed out in my meta review, I believe the amount of clarification required is extensive and this paper will at least need a significant revision before it can be accepted.

---

### Decision · Program_Chairs · 2025-01-22

Reject